# Pre-Training on In Vitro and Fine-Tuning on Patient-Derived Data Improves Deep Neural Networks for Anti-Cancer Drug-Sensitivity Prediction

**DOI:** 10.3390/cancers14163950

**Published:** 2022-08-16

**Authors:** Paul Prasse, Pascal Iversen, Matthias Lienhard, Kristina Thedinga, Ralf Herwig, Tobias Scheffer

**Affiliations:** 1Department of Computer Science, University of Potsdam, 14476 Potsdam, Germany; 2Department of Computational Molecular Biology, Max Planck Institute for Molecular Genetics, 14195 Berlin, Germany

**Keywords:** deep neural networks, drug-sensitivity prediction, anti-cancer drugs

## Abstract

**Simple Summary:**

Cancer cell lines vary greatly from one another because each one underwent a combination of random mutations. Therefore, the effectiveness of anti-cancer drugs also varies across cancer cell lines. In order to understand which drugs are effective against which cancer cells, researchers expose cultivated cell lines to drug candidates. However, these results are not entirely realistic, mostly because cultivated cells live in a two-dimensional in vitro environment without interacting with other types of cells. There are more realistic ways to test drug effectiveness—e.g., organoids that are 3D-printed from tumor cells—but since these processes are more complex, much fewer data are available. We studied an approach in which a neural network is first trained on the wealth of available in vitro drug-sensitivity data, before being fine-tuned on smaller but more realistic databases. We found that this training procedure improves the neural network’s ability to predict how effective a particular drug will be for a given tumor cell line. Such neural networks can serve as a tool, both for personalized treatment of cancer and for drug development.

**Abstract:**

Large-scale databases that report the inhibitory capacities of many combinations of candidate drug compounds and cultivated cancer cell lines have driven the development of preclinical drug-sensitivity models based on machine learning. However, cultivated cell lines have devolved from human cancer cells over years or even decades under selective pressure in culture conditions. Moreover, models that have been trained on in vitro data cannot account for interactions with other types of cells. Drug-response data that are based on patient-derived cell cultures, xenografts, and organoids, on the other hand, are not available in the quantities that are needed to train high-capacity machine-learning models. We found that pre-training deep neural network models of drug sensitivity on in vitro drug-sensitivity databases before fine-tuning the model parameters on patient-derived data improves the models’ accuracy and improves the biological plausibility of the features, compared to training only on patient-derived data. From our experiments, we can conclude that pre-trained models outperform models that have been trained on the target domains in the vast majority of cases.

## 1. Introduction

The treatment of cancer heavily relies on standard-of-care therapies that fail to address the diverse nature of the disease. Being caused by a combination of genetic mutations, the sensitivity to drug compounds varies greatly across cancer cell lines, which results in a vast range of therapeutic outcomes for seemingly similar clinical presentations. Genomic alterations and the transcriptomes of cancer cells, in combination with the biochemical mechanisms underlying the administered drugs, are among the factors that determine the diverse outcomes of cancer therapies [1]. Driven by advances in genomic testing, precision oncology aims at providing the best therapy for individual patients, given all available information, including their genomic and transcriptomic tumor profiles.

Increasingly, data-driven machine-learning approaches are used to facilitate precision oncology [2]. Machine-learning approaches rely on large-scale drug-sensitivity screening data, such as the Genomics of Drug Sensitivity in Cancer Database (GDSC) [3,4] and the Cancer Cell Line Encyclopedia (CCLE) [5]. These data are generated by exposing cultivated tumor cell lines to a variety of anti-cancer compounds and measuring the survival rates of the cancer cells as a function of the drug concentration. A number of well-known drug-sensitivity models have been developed with these in vitro databases [6,7], including the PaccMann model [7], which predicts the inhibitory concentration of a pair of drug and cell lines based on a tokenized SMILES-string representation of the drug molecule and transcriptomic features.

Cultivated cancer cell lines are easy to handle and therefore useful for the creation of large-scale drug-sensitivity databases. However, since these cell lines have devolved from human cancer cells over years, and in some cases, decades of selective pressure in culture conditions, they may no longer faithfully represent the molecular characteristics of primary patients’ tumors [8,9]. To address this issue, several patient-derived model systems for cancer have been developed, with different levels of complexity. Patient-derived ex vivo cell cultures [10] (PDCs) provide screenable cell model systems more similar to the primary patients’ tumors. While being closer approximations to clinical presentations of cancer than the cell lines that have been cultivated over many generations, ex vivo cell cultures still lack cell–cell interactions in a three-dimensional environment. Next, for patient-derived xenografts [11] (PDXs), the tumor is implanted into living animals, and thus PDXs allow drug-sensitivity screening in vivo, albeit in a non-human tissue environment. Finally, patient-derived organoids [12] (PDOs) are self-organizing 3D cultures containing different cell types, resembling essential aspects of the organs that host the primary patient tumors. PDOs facilitate in vitro efficacy screening of multiple drugs in a multicellular tissue-like environment.

While better representing clinical tumor cases, the downside of patient-derived cell cultures, xenografts, and organoids lies in the higher complexity of the processes. In consequence, ex vivo, xenograft, and organoid drug-sensitivity data are available at a much smaller scale that is not sufficient to train high-capacity neural networks that represent the state of the art in machine learning [7,13].

In this study, we explored a transfer-learning approach that exploits the abundance of in vitro drug-sensitivity data to pre-train models which are then fine-tuned on drug-sensitivity data of patient-derived cell lines. In order to obtain results that are not just specific for a particular machine-learning model, we compared the performance of this transfer-learning approach to the state-of-the-art PaccMann model [7], the best model of Zhu et al. [6], and a relatively simple convolutional neural network architecture as a baseline model. We evaluated the models in terms of their performance in two different use cases. In the use case of *precision oncology*, the sensitivity of a known panel of drugs has to be predicted for a new, previously unseen cell line at hand; this setting is also referred to as *cell cold start*. In the use case of *drug development*, by contrast, the sensitivity of a known panel of cell lines to a new, previously unseen drug molecule is predicted; this setting is also referred to as *drug cold start*.

Prior work on transfer learning in this domain by Zhu et al. [6] has demonstrated that it is possible to pre-train deep-learning drug-sensitivity prediction models on a large data set, but their work purely focused on cultivated cell lines. By contrast, we also consider patient-derived cell lines as the target domain. Ma et al. [14] studied transfer learning from in vitro data to PDO and xenograft target domains. They followed an approach of training a separate neural-network model for each drug compound that only receives cell-line information as input. This approach does not allow the possibility of generalization to unseen drugs, which is crucial for the use case of drug development.

## 2. System and Methods

In this section, we introduce the data sets, the formal problem setting, the experimental setting, and the neural-network models.

### 2.1. Data

All data sets that we used in this study report gene expression profiles for a number of cancer samples, and drug-sensitivity measurements for a set of drugs. They differ in size, type of cancer model, and methods to measure gene expression and drug sensitivity.

We pre-trained all models on the cancer-specific data from the *Genomics of Drug Sensitivity in Cancer project (GDSC)* [3,4], because with 958 cell lines and 282 drugs, this is the largest publicly available in vitro drug-sensitivity database.

Four other data sets were used for model tuning and evaluation. They served as the target domain, representing pre-clinical cancer models with deviating properties and different approximations of the primary patient tumors. The first target data set is the *Cancer Cell Line Encyclopedia (CCLE)* [5] with 24 drugs, of which 7 overlap with GDSC (see Table 1). Even though CCLE, like GDSC, is based on cultivated cell lines, differences in the techniques that are used to measure RNA expression levels impede the direct application of the GDSC-trained model. Next, the *Beat Acute Myeloid Leukaemia (Beat AML)* data set [15] reports drug-sensitivity data for 213 AML patient-derived cell cultures (PDCs) and 109 drugs.

Furthermore, we fine-tuned the neural networks with a *lung cancer xenografts* data set (PDXs) [16]. For this study, tumor samples from 19 lung cancer patients were implanted into mice and treated with 3 different cancer drugs. In contrast to in vitro data sets, drugs were administered the same way as in clinics, and sensitivity was measured by the relative size of the tumor in treated xenografts compared to untreated (control) xenografts. Finally, the *Pancreatic Cancer Patient-derived Organoid (PDO)* data set [17] quantified the drug sensitivities of 25 drugs via the area under the dose–response curve (AUC) in 44 PDOs that form epithelial structures and have been shown to closely resemble the transcriptomic profiles and treatment responses of the hosts. Table 1 shows the number of cell lines, the number of drugs, and the overlap of drugs with the GDSC data in each data set.

### 2.2. Problem Setting and Performance Metrics

We studied two variants of problem settings that represent the use cases of *precision oncology* and *drug development*, respectively.

The goal of *precision medicine* is to predict the effect of a range of available drugs for a given, previously unseen, tumor case. Drug sensitivity is measured in terms of the inhibitory concentration IC50 for the CCLE data set, in terms of the AUC for the Beat AML and the PDO data sets, and in terms of tumor growth for the Xenografts data set. For precision medicine, we evaluated predictive performance as the mean value, across cell lines, of the *Pearson correlation* between predicted and measured drug sensitivity for all drugs. This metric was derived for tumor cases not used in training, corresponding to the *cell cold-start problem*. The average correlation between observed and predicted inhibitory concentration over all cell lines measures a model’s ability to rank drugs according to their inhibitory capacities for a new tumor case—i.e., cell line—at hand.

Note that the Pearson correlation is translation and scale invariant. A predictive model that scales the inhibitory concentrations incorrectly while sorting compounds perfectly according to their true inhibitory concentrations can achieve a perfect Pearson correlation. This corresponds exactly to the goal of identifying the best candidate drugs for a given cell line. We also report the mean squared error (MSE) to highlight the mean deviations of the predictions from the ground truth.

For *drug development*, we evaluated the mean value, across drugs, of the Pearson correlation between predicted and measured drug sensitivity for all cell lines. This calculation was performed for drug compounds that were present in the training data, corresponding to the *drug cold-start problem*. The average correlation over all drugs quantifies a model’s ability to identify cell lines for which a novel candidate compound may be an effective inhibitor.

Again, the Pearson correlation, as an evaluation metric, quantifies the model’s ability to identify the most promising cell lines for a give drug compound while abstracting from the scale and translation of the output values. Additionally, we report the mean squared error (MSE) between predicted and ground-truth inhibitory concentrations.

We judged the statistical significance of differences in the performances of models with two-sided paired Student’s *t*-tests with a *p* value threshold of 0.05.

### 2.3. Experimental Setting

We used the GDSC database for pre-training and each of the remaining data sets of Table 1 as the target domain. We employed 10-fold cross-validation to evaluate all models. For precision oncology (cell cold-start), we split the target data along cell lines such that no cell line occurred in the training and test portions at the same time. For the CCLE data set in which cell lines partly overlap with GDSC, we removed target cell lines from the GDSC database prior to training to avoid data leakage from the training to the testing data. For the drug-development use case (drug cold-start), we split the target data by drug and removed drugs that occur in the target data from GDSC training data.

In each cross-validation step, all models were first pre-trained with GDSC data and then fine-tuned by training on the training portion of the target data set. To investigate the impact of the size of the target data set, we repeated fine-tuning on fractions of the training data. The performances of these pre-trained models were compared to those of models trained on the target data set alone. We trained the models with early stopping (patience of 5): we used 90% of the training data for training and 10% of the training data for early-stopping validation.

Models were trained on two different data sets (one for pre-training and the other for fine-tuning), which is not straightforward because the transcriptomic features exhibit a covariate shift due to different RNA measuring methods. We used the data harmonization technique COMBAT, which is based on a linear mixed model to transform the RNA measurements [18]. By applying the COMBAT method, we ensured that the distributions of RNA measurements for the GDSC and the target data set are similar. To avoid data leakage, the COMBAT method was only fitted on the pre-training and training data and then applied to the testing data.

We trained all models using the Keras [19] and Tensorflow [20] libraries and the NVidia CUDA platform on a NVIDIA A100-SXM4-40GB GPU. We implemented the evaluation framework using the scikit-learn [21] machine-learning package. The code can be found online at https://github.com/prassepaul/mlmed_transfer_learning (accessed on 14 July 2022).

### 2.4. Drug-Sensitivity Models

All investigated models were pre-trained to predict the IC50s on the GDSC data. For fine-tuning on one of the target domains, the output layer was replaced with a layer that produces the appropriate target output metric (see Figure 1A). All models use transcriptomic features to characterize the cancer cases, and either sets of molecular features or SMILES strings representing the drugs. A SMILES string is a one-dimensional representation of a molecule, based on a walk in the atom-bond network structure. In our evaluation protocol, we compared three different models with each other: PaccMann [7], tDNN [6], and a convolutional neural-network architecture (Conv NN).

#### 2.4.1. PaccMann

PaccMann [7] is a state-of-the-art model that predicts the effectiveness of drugs on cancer cell lines (see Figure 1B). PaccMann incorporates prior knowledge about drug-target information and protein–protein interactions to restrict the number of gene expression features used by the neural network to relevant genes. Using network propagation on a protein–protein interaction network [22], the 20 highest-scoring genes for each compound are identified, which results in a total set of 2089 relevant target genes, for which expression levels are used in the model. PaccMann uses attention-based network modules to encode both the tokenized SMILES string and the gene expression data. The network parameters are trained to minimize the mean squared error.

#### 2.4.2. tDNN

The tDNN model uses two sub-networks [6] of three hidden dense layers each. The first sub-network processes the gene expression, the second one receives the drug descriptors as input (see Figure 1C). The outputs of the two sub-networks are concatenated and then passed to four hidden dense layers before output. In each hidden layer, the number of nodes is reduced by 50%, starting with 1000 nodes after the input. Parameters are trained to minimize the mean-squared error.

This model uses expression levels for a set of around 1900 genes, including genes that represent cellular transcriptomic changes identified by the LINCS project [23], cancer-related genes collected from OncoKB [24], and cancer-related genes from GDSC [4]. The model uses approximately 1600 descriptive features for the drugs that are computed using the commercial Dragon software package. Features include numeric features based on the molecular structure, such as the simplest atom types, functional groups and fragment counts, topological and geometrical descriptors, estimations of molecular properties, and drug-like and lead-like indices. We reconstructed these features using the open-source Python libraries RDKit (https://github.com/rdkit/rdkit, accessed on 14 July 2022) and Mordred (https://github.com/mordred-descriptor/mordred, accessed on 14 July 2022).

#### 2.4.3. Conv NN

The third neural-network architecture under investigation (Conv NN) consists of two sub-networks. The first processes the tokenized SMILES string using an embedding layer, followed by three layers of one-dimensional convolutions and max-pooling to extract important features from the molecule structure. The result of the first sub-network is then converted to a flat vector and concatenated with the output of the second sub-network. The second sub-network consists of a fully connected layer that processes the gene expression input. The concatenation of the two sub-networks is then fed into three fully connected layers with batch normalization before the output layer (see Figure 1D). The network parameters are trained to minimize the mean-squared error.

### 2.5. Plausibility Evaluation of Model Predictions

In order to assess the reliability of the different models, we investigated the biological plausibility of the high-ranked features (genes) identified by the models. We used the *integrated gradients* method to highlight the most important features for a given input sequence [25] using an all-zeros baseline. Integrated gradients can compute the contributions of genes when classifying a drug–PDO pair. Subsequently, we performed over-representation analysis for the top-ranked genes on pre-annotated pathway gene sets as provided by the ConsensusPathDB [26] using Fisher’s exact test. Pathway *p*-values were corrected according to the Benjamini–Hochberg method.

## 3. Results

This section reports the results and findings of our evaluation. Table 2 shows the raw experimental results.

### 3.1. Transfer Learning Facilitates Precision Oncology

In the context of precision oncology (cell cold-start), Table 2 shows a common pattern for all investigated models: When up to 1000 training examples from the target domain are available, pre-training significantly improves the predictions for precision oncology compared to training from scratch. Results marked with an asterisk (“*”) indicate significant improvements (p<0.05, based on a two-sided Student’s *t*-test). When using 100 training examples, the pre-trained models always performed better for the CCLE, BeatAML, and Organoid data when evaluating the models based on the mean squared error (MSE). For the CCLE data, MSEs between 0.172 and 0.532 were achieved using models trained from scratch, versus MSEs between 0.11 and 0.136 for pre-trained models. For the Beat AML data, MSEs between 0.088 and 0.561 were achieved using models trained from scratch, versus MSEs between 0.056 and 0.07 for pre-trained models. For the Organoid data, MSEs between 0.028 and 0.524 were achieved, versus MSEs between 0.02 and 0.032 for pre-trained models.

When more than 1000 tuning examples from the target domain were available, the difference between pre-trained and trained-from-scratch models became insignificant in most cases. There was no case in which a pre-trained model was significantly worse than the corresponding model trained from scratch.

In most cases, even the pre-trained model without any fine-tuning was already a reasonable baseline; the model trained from scratch took roughly between 100 and 500 training examples from the target domain to outperform the corresponding model that had only been trained on GDSC. When less than 100 training examples from the target domain were available, fine-tuning was often detrimental for the GDSC-trained model. This is plausible because the subsequent gradient-update steps can over-fit the initially reasonable models to extremely small tuning samples. For the Xenograft data with only 19 patients and 3 drugs, the GDSC-trained models are more accurate than all fine-tuned models.

### 3.2. Transfer Learning Facilitates Drug Development

In the drug-development use case (drug cold-start), Table 2 reports generally lower Pearson correlation values than for precision oncology. This is plausible for two reasons: Firstly, the training data contain roughly three times as many cell lines as drugs; generalization across drugs is therefore based on fewer data than generalization across cell lines. Secondly, we believe that the transcriptomic features facilitate generalization across cell lines because cell lines with similar transcriptomes are likely to share similar drug sensitivities. By contrast, compounds with similar SMILES strings, or in the case of the tDNN model, similar features calculated by the Dragon software package, do not necessarily share similar pharmaceutic effects to the same extent.

In contrast to precision oncology, we also found the performance benefit of pre-trained models in drug development settings to be generally less dependent on the number of drug–cell-line pairs. Instead, we observed an increase in predictive performance by pre-training over the complete range of sample sizes, similarly for all considered prediction models and all target data sets. In many cases, the improvements by pre-training are significant. The results for the extremely small in vivo xenograft data set are not conclusive.

When using 100 training examples, the pre-trained models always performed better on the CCLE, BeatAML, and Organoid data when evaluated using the MSE. For the CCLE data, MSEs between 0.231 and 0.588 were achieved using models trained from scratch, versus MSEs between 0.197 and 0.222 for pre-trained models. For the Beat AML data, MSEs between 0.079 and 0.478 were achieved using models trained from scratch, versus MSEs between 0.061 and 0.068 for pre-trained models. For the Organoid data, MSEs between 0.044 and 0.498 were achieved, versus MSEs between 0.043 and 0.053 for pre-trained models.

### 3.3. Transfer Learning Improves the Plausibility of Features

We compare the feature importance values attributed to the genes between pre-trained and scratch models (cf. Methods) and exemplify this with the pancreatic cancer PDO data set [17] for the PaccMann method [7]; cf. Table 2.

For each of the 1093 drug–PDO pairs, the pre-trained and scratch models assigned a feature importance value to all of the genes. In a first step, we extracted for each gene the overall sum of feature importance values across all drug–PDO pairs. From the total of 2029 attended genes, we were interested in the top-ranked genes from pre-trained and scratch models, and we observed that these genes are fairly different, there being only 24 overlapping genes (12%; Figure 2A). This suggests that pre-training leads to a different set of genes being selected by the model compared to scratch predictions. This observation holds as well with other thresholds. Over-representation analysis with pre-annotated pathways resulted in 62 enriched pathways for the 200 top-ranked genes of the scratch model, in contrast to 120 enriched pathways for the 200 top-ranked genes of the pre-trained model (overlap of 48 pathways), which indicates much higher biological information content of the pre-trained model genes. Moreover, Figure 2B shows that the levels of enrichment and over-representation of pre-trained model genes are far more significant than those of scratch model genes (p<0.0001), indicating a more plausible feature selection.

Next, we repeated the analysis by extracting the top 200 ranked genes of the scratch and pre-trained models for a specific drug. We chose the PARP inhibitor olaparib, since this is an approved targeted therapy for pancreatic cancer [27], and summarize the feature importance values of the 44 olaparib–PDO pairs. As in the overall analysis, the selected genes are very distinct and overlap in 18 genes only (9%; Figure 2C). Furthermore, over-representation analysis with the 200 top-ranked genes from the scratch model resulted in 84 enriched pathways, in contrast to 126 enriched pathways for the 200 top-ranked genes from the pre-trained model (overlap 54 pathways). As previously, the level of enrichment with the pre-trained model genes was far more significant (p<0.001; Figure 2B). Olaparib inhibits the poly(ADP-ribose) polymerase 1 and 2 genes (PARP1 and PARP2), which play an important role in DNA damage repair [28], and thus targets genome integrity. We found that the top-ranked genes from the pre-trained model enrich a couple of relevant pathways related to DNA repair and replication, for example, "cell cycle" (*q*-value $=3.26\times 10^{-15}$), "cellular senescence" ($q=1.12\times 10^{-10}$), "Apoptosis" ($q=2.79\times 10^{-06}$), "transcriptional regulation by TP53" ($1.03\times 10^{-09}$), "E2F transcription factor network" ($4.73\times 10^{-10}$), and "MAPK signaling" ($3.81\times 10^{-08}$). These pathways were far less enriched by the top-ranked genes from the scratch model (Figure 2D).

### 3.4. Transfer Learning Improves Predictions towards Established Therapies in Pancreatic Cancer

The PDOs used by Tiriac et al. [17] represent a wide range of therapeutically relevant pancreatic cancer samples. We observe that the accuracies of pre-trained models are higher than those of models trained from scratch (see Table 2). This holds particularly for established therapies. The major chemotherapies for pancreatic cancer are gemcitabine/paclitaxel and FOLFIRINOX, which is composed of several components such as 5-FU, irinotecan (SN-38), and oxaliplatin [27,29]. Figure 3A shows the accuracies of the pre-trained and scratch models, measured in absolute differences of the predicted and measured AUCs for all drug–PDO pairs, in the case of the PaccMann model. It is observable that for most drugs—except gemcitabine—the pre-trained model predicted the AUC more accurately than the scratch model without pre-training. This increase in prediction accuracy was significant in the cases of oxaliplatin (p=0.00110); SN-38 ($p<10^{-4}$), the active metabolite of irinotecan; and the targeted therapy olaparib (p<0.01) that has initially been approved for breast and ovarian cancers with BRCA mutations.

The 44 PDOs can be further subdivided into molecular subtypes, a basal subtype (13 PDOs), and a classical subtype (31) [30]. Improvement is also observable concerning these subtypes; for example, the targeted drug olaparib has better predicted AUCs in both subgroups when comparing the pre-trained model versions with the scratch models (see Figure 3B). For the classical subgroup, this difference in prediction accuracy is significant (p=0.0435), and for the basal subgroup, the difference is weakly significant (p=0.0642).

### 3.5. Transfer Learning Offers Novel Treatment Options

Since AUCs indicate sensitivity and resistance, respectively, of the PDO to the drug, we are interested in whether the pre-trained model can be used for predicting individual responses of the PDOs to different drugs. Since drugs have different sensitivity ranges, it is difficult to directly compare drugs based on AUCs. We, therefore, denote for each drug the *z*-scores for the 44 PDOs. Figure 3C shows the drugs with the five lowest *z*-scores for selected PDOs. Consistent results could be derived for, for example, the PDO hF28: the second-lowest AUC *z*-score for this PDO has been assigned to the drug KRAS-G12C, an inhibitor of the KRAS mutant protein (*z*-score = −1.49). This is consistent with the mutational status of the PDO; it has been documented that hF28 is KRAS-mutated (G12D) with an allele frequency of 46%. Furthermore, Tiriac et al. [17] presented intensive patient treatment history for one patient (PDO hF2) that was found resistant for paclitaxel and weakly responsive for oxaliplatin and 5-FU so that the patient had to be treated with second-line therapies. The results of the pre-trained model for this PDO show overall poor responses for all drugs under study. Although paclitaxel has the lowest *z*-score for this PDO among all drugs in the pre-trained model (0.67), the score is not significant, and also the scores for 5-FU (1.33) and oxaliplatin (2.46) indicate resistance rather than sensitivity.

Established first-line therapies (see above) are among the top five predicted drugs for 30 of the 44 PDOs (68%) and among the top 10 predicted drugs for 39 (89%). For five PDOs, the model predicted other options (hF3, hF32, hM19D, hT96, hM21F). All these five PDOs had KRAS mutations; for one PDO (hF3), the mutated-KRAS inhibitor KRAS-G12C was among the top five predictions, and for the other four PDOs, afatinib was among the top five predictions. This result shows agreement with the literature: a recent clinical phase-I study was reported for the application of afatinib (together with selumetinib) against KRAS-mutated tumors in patients suffering from pancreatic and other cancers [31]. We conclude that it is possible to deduce plausible treatment options from the pre-trained prediction model.

## 4. Discussion

We have explored a method for training pre-clinical anti-cancer drug-sensitivity models that are based on pre-training on in vitro drug-sensitivity data and fine-tuning on drug-sensitivity data based on patient-derived cell cultures, xenografts, or organoid models. We have explored pre-training of state-of-the-art model architectures [6,7] and reasonable baseline methods.

From our experiments, we can conclude that pre-trained models outperform models that have been trained on the target domain in the vast majority of cases. This is particularly true for target data sets with up to 1000 training instances from the target domain. When fewer than 100 tuning instances from the target domain are available, fine-tuning can even be detrimental to the pre-trained model. Moreover, there was no case in which a model that had only been trained on data from the target domain significantly outperformed the pre-trained model. Perhaps surprisingly, no dominant model architecture emerged in our experiments; the simple convolutional neural network appears largely on par with PaccMann and tDNN. This finding underscores the overwhelming importance of training data, which can easily outweigh improvements in model architectures.

For the xenograft data, fine-tuning did not achieve a significant improvement over the GDSC-trained model. This may have been due to the small sample causing the initially reasonable pre-trained model to over-fit.

By contrast, we have shown that pre-training improves the prediction model for PDOs. PDOs have recently gained popularity in drug testing, since they map tumor heterogeneity while preserving the molecular characteristics of the original cancer. Their usage in personalized medicine is currently under investigation by many research groups [17,32]. We have explored how the large body of cell line drug-sensitivity data can be used to aid in this development. We showed that pre-training with in vitro data improves prediction error in the vast majority of PDO–drug pairs, particularly for approved drugs (Figure 3). In addition, we showed how the prediction models can be used to generate plausible drug recommendations for individual PDOs.

DNN predictions have been compared, on the one hand, according to their ability to predict drug-sensitivity values (Table 2). On the other hand, it is of interest which features gain importance based on their gene expression patterns in the prediction processes of the different models. In terms of selected features, we conclude that the computational models with pre-training identify features with higher biological plausibility than without, which could be seen with pathway over-representation analysis (Figure 2). This holds in particular with respect to specific drugs, which has been exemplified with the inhibitor olaparib. Features selected by pre-training seem to better describe the mechanism of action of the specific drug, thereby making it biologically more plausible. This not only leads to a better prediction but may also indicate the advantages of pre-training for identifying suitable therapy markers.

We found that the precision oncology use case (cell cold-start) is generally easier than drug development (drug cold-start). We believe that this discrepancy is caused by the number of drugs in the training data being smaller than the number of cell lines, and by the transcriptomic features facilitating generalization better than the SMILES representation, and in the case of the tDNN model, numeric features calculated by the Dragon software, of drug compounds. Further research on the representations of drugs to uniquely characterize their 3D structures or their biochemical interactions may improve the model’s ability to generalize across drugs, and thereby the model’s performance in the drug development use case.

## 5. Conclusions

In this study, we have shown that DNN models can effectively been used to predict drug-sensitivity values in pre-clinical models. To circumvent the relatively small number of drug–cell system pairs, we explored a transfer-learning approach that exploits the large body of in vitro drug-sensitivity data to pre-train DNN models which were then fine-tuned on drug-sensitivity data of patient-derived systems. We observed that pre-training improves prediction accuracy in most of the applications, in particular, in a range of 10–1000 training samples. We further demonstrated that feature importance values from pre-trained models identified genes with higher biological plausibility, as shown with over-representation analysis of biological pathways that monitor molecular responses to targeted drugs. In summary, our transfer learning approach offers an effective use of large drug databases to optimize drug-sensitivity predictions, even in small-scale patient-derived cell system studies.

## Figures and Tables

**Figure 1 cancers-14-03950-f001:**
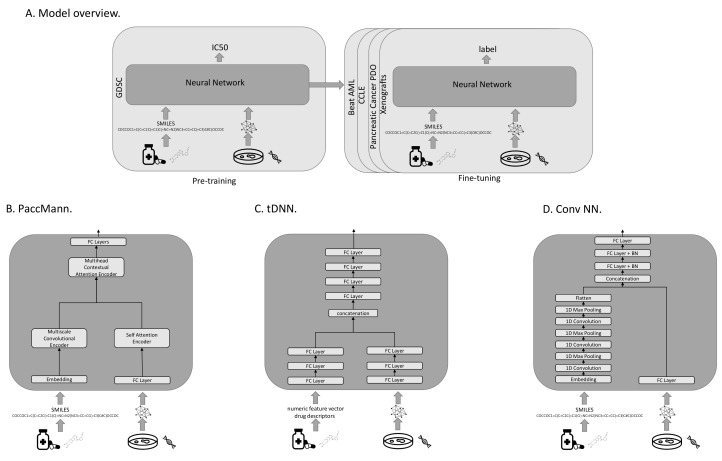
Overall model architecture (**A**) and models. Model architectures for the PaccMann model (**B**), the tDNN model (**C**), and the Conv NN (**D**).

**Figure 2 cancers-14-03950-f002:**
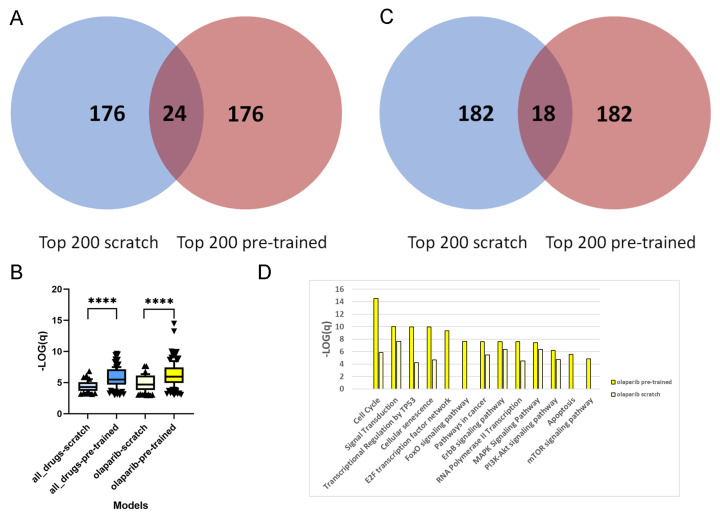
Feature importance analysis. (**A**) Venn diagram of top 200 features for the model trained from scratch (blue) and the pre-trained (red) PaccMann model on the PDO data set. Feature importance values were averaged over all drug–PDO pairs. (**B**) Box-plots of pathway enrichment scores of the top-ranked 200 genes derived from feature importance values of the pre-trained model and the model trained from scratch for all drugs and specifically for the drug olaparib (x-axis). The y-axis denotes the $-\log_{10}$ of the adjusted enrichment *p*-values. Boxes denote the 10–90% ranges of pathway enrichment scores. Pathway gene sets were taken from the ConsensusPathDB resource and were judged as significantly enriched if Fisher’s exact test resulted in a *p*-value of below 0.001, and if the pathway shared at least 10 genes with the 200 top-ranked genes. Sizes of samples were: all-drugs-scratch: 62 pathways; all-drugs-pre-trained: 120; olaparib-scratch: 84; olaparib-pre-trained: 126. On top of each model pair, the result of an unpaired, two-sided Mann–Whitney test is displayed (****: p<0.0001). (**C**) Venn diagram of top 200 features from the scratch (blue) and pre-trained (red) PaccMann models and the PDO data set. Feature importances were averaged over the olaparib–PDO pairs. (**D**) Bar plot of selected enrichment pathway scores ($-\log_{10}$ of adjusted enrichment *p*-value, y-axis) for the 200 top-ranked genes from the scratch (light yellow) and pre-trained (strong yellow) PaccMann models using the feature importance values for all olaparib–PDO pairs; x-axis: pathways related to DNA damage repair and replication.

**Figure 3 cancers-14-03950-f003:**
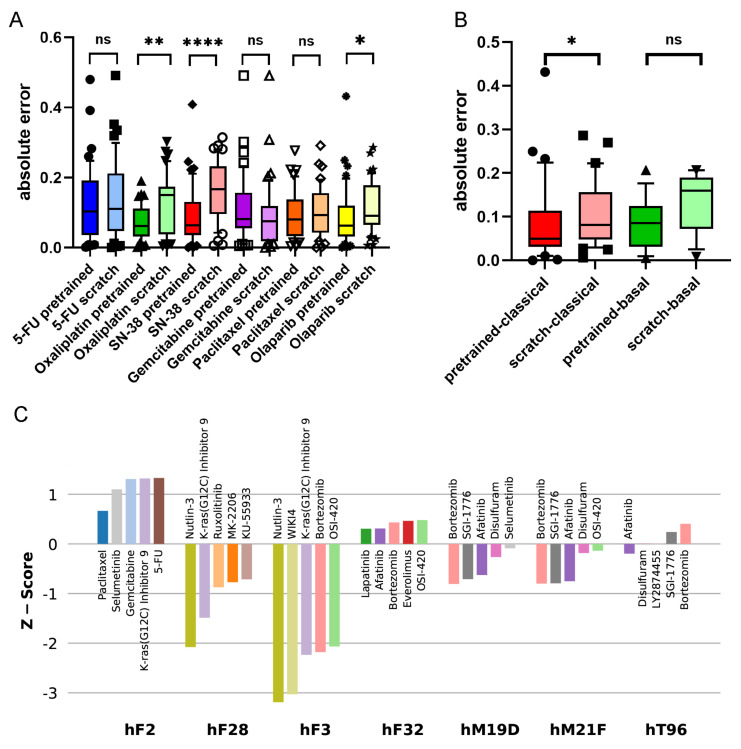
Effect of pre-training for PDO–drug AUC predictions. (**A**) Absolute differences in predicted and ground truth AUCs (y-axis) for standard therapies with pre-trained and models trained from scratch (x-axis). Boxes show the 10–90% ranges of absolute errors for the different PDOs; the line within each box depicts the median value. Bright colors show the pre-trained models; lighter colors show the trained-from-scratch models. On top of each model pair, the result of an unpaired, two-sided Mann–Whitney test is displayed (****: p<0.0001; **: p<0.01; *: p<0.05; ns: not significant). (**B**) Absolute differences in predicted and ground truth AUCs (y-axis) for the targeted drug olaparib when PDOs are grouped according to basal and classical subtypes. On top of each model pair, the result of an unpaired, two-sided Mann–Whitney test is displayed (****: p<0.0001; **: p<0.01; *: p<0.05; ns: not significant). (**C**) *Z*-scores (y-axis) of the 5 best drug predictions for selected PDOs (x-axis). Different drugs are indicated by colored bars.

**Table 1 cancers-14-03950-t001:** Statistics of used data sets.

Data Set	Type	# Cell Lines/	# Drugs	# Samples	Drug-Overlap
		Patients			with GDSC
GDSC [3,4]	Cultivated cell lines	958	282	250,625	–
CCLE [5]	Cultivated cell lines	472	24	10,924	7
Beat AML [15]	Patient-derived cell culture	213	109	18,062	31
Xenografts [16]	Patient-derived xenograft	19	3	120	2
PDO [17]	Patient-derived organoid	44	25	1093	13

**Table 2 cancers-14-03950-t002:** Average Pearson correlations for different models and the different data sets. For results marked “*”—the correlation of the pre-trained model is significantly higher (p<0.05) than that of the same model trained from scratch. Results highlighted in bold mark the best performing model regarding the model trained from scratch and its pre-trained version.

Data		# Train	Conv NN	Conv NN	PaccMann	PaccMann	tDNN	tDNN
			Scratch	Pre-Trained	Scratch	Pre-Trained	Scratch	Pre-Trained
CCLE data	precision oncology	0	–	0.615 ± 0.009	–	0.575 ± 0.006	–	0.614 ± 0.006
		10	0.053 ± 0.023	**0.437 ± 0.041** *	0.039 ± 0.05	**0.538 ± 0.026** *	0.004 ± 0.023	**0.473 ± 0.042** *
		50	0.235 ± 0.033	**0.676 ± 0.016** *	−0.018 ± 0.054	**0.634 ± 0.017** *	0.451 ± 0.024	**0.618 ± 0.021** *
		100	0.384 ± 0.026	**0.714 ± 0.005** *	0.022 ± 0.049	**0.666 ± 0.011** *	0.586 ± 0.018	**0.667 ± 0.011** *
		500	0.717 ± 0.007	**0.776 ± 0.004** *	0.379 ± 0.042	**0.725 ± 0.007** *	**0.749 ± 0.01**	0.74 ± 0.009
		1000	0.686 ± 0.022	**0.758 ± 0.011** *	0.353 ± 0.049	**0.741 ± 0.008** *	**0.764 ± 0.005**	0.759 ± 0.005
		5000	0.775 ± 0.006	**0.78 ± 0.005**	**0.751 ± 0.006**	0.729 ± 0.006	**0.781 ± 0.005**	0.78 ± 0.006
		all	0.777 ± 0.008	**0.781 ± 0.006**	**0.757 ± 0.007**	0.745 ± 0.006	**0.789 ± 0.004**	0.781 ± 0.006
	drug development	0	–	0.155 ± 0.027	–	0.072 ± 0.018	–	0.209 ± 0.036
		10	−0.005 ± 0.021	**0.044 ± 0.028**	−0.018 ± 0.011	**0.1 ± 0.019** *	0.051 ± 0.025	**0.108 ± 0.025**
		50	−0.019 ± 0.02	**0.054 ± 0.029**	0.008 ± 0.018	**0.067 ± 0.025**	0.058 ± 0.02	**0.101 ± 0.024**
		100	0.013 ± 0.028	**0.08 ± 0.022**	0.01 ± 0.011	**0.061 ± 0.024**	0.066 ± 0.014	**0.106 ± 0.024**
		500	0.043 ± 0.019	**0.139 ± 0.018** *	−0.002 ± 0.015	**0.087 ± 0.014** *	0.126 ± 0.013	**0.172 ± 0.018**
		1000	0.063 ± 0.029	**0.158 ± 0.009** *	0.029 ± 0.009	**0.069 ± 0.014***	0.15 ± 0.015	**0.174 ± 0.013**
		5000	0.282 ± 0.021	**0.293 ± 0.015**	**0.237 ± 0.02**	0.181 ± 0.016	**0.302 ± 0.021**	0.284 ± 0.024
		all	0.29 ± 0.019	**0.293 ± 0.014**	**0.255 ± 0.021**	0.22 ± 0.011	0.328 ± 0.021	**0.338 ± 0.028**
Beat AML data (PDCs)	precision oncology	0	–	0.229 ± 0.011	–	0.258 ± 0.012	–	0.252 ± 0.01
		10	−0.014 ± 0.014	**0.182 ± 0.018** *	−0.007 ± 0.026	**0.251 ± 0.022** *	−0.08 ± 0.02	**0.211 ± 0.028** *
		50	0.015 ± 0.018	**0.315 ± 0.031** *	0.003 ± 0.026	**0.275 ± 0.016** *	−0.035 ± 0.018	**0.285 ± 0.022** *
		100	0.049 ± 0.02	**0.346 ± 0.037** *	0.03 ± 0.03	**0.305 ± 0.015** *	0.052 ± 0.036	**0.319 ± 0.03** *
		500	0.26 ± 0.024	**0.564 ± 0.011** *	0.023 ± 0.023	**0.479 ± 0.015** *	0.332 ± 0.07	**0.506 ± 0.013** *
		1000	0.135 ± 0.032	**0.524 ± 0.035** *	0.094 ± 0.018	**0.534 ± 0.015** *	**0.61 ± 0.017**	0.595 ± 0.013
		5000	**0.676 ± 0.008**	0.674 ± 0.011	0.638 ± 0.026	**0.643 ± 0.008**	**0.685 ± 0.01**	0.67 ± 0.012
		10,000	**0.703 ± 0.01**	0.7 ± 0.01	**0.677 ± 0.008**	0.652 ± 0.009	**0.697 ± 0.009**	0.693 ± 0.01
		all	0.693 ± 0.01	**0.695 ± 0.012**	**0.685 ± 0.01**	0.676 ± 0.009	**0.706 ± 0.012**	0.696 ± 0.009
	drug development	0	–	0.074 ± 0.012	–	0.008 ± 0.015	–	0.051 ± 0.013
		10	−0.005 ± 0.016	**0.018 ± 0.012**	−0.004 ± 0.007	**0.018 ± 0.015**	0.02 ± 0.018	**0.059 ± 0.02**
		50	0.007 ± 0.014	**0.048 ± 0.027**	0.019 ± 0.005	**0.023 ± 0.01**	0.044 ± 0.017	**0.077 ± 0.016**
		100	−0.002 ± 0.017	**0.114 ± 0.022** *	0.005 ± 0.009	**0.031 ± 0.007**	0.089 ± 0.015	**0.092 ± 0.017**
		500	0.041 ± 0.01	**0.243 ± 0.013** *	0.003 ± 0.007	**0.108 ± 0.011** *	0.182 ± 0.021	**0.204 ± 0.015**
		1000	0.087 ± 0.021	**0.225 ± 0.024** *	0.006 ± 0.01	**0.128 ± 0.014** *	**0.293 ± 0.015**	0.292 ± 0.016
		5000	0.261 ± 0.027	**0.368 ± 0.014** *	**0.351 ± 0.013**	0.273 ± 0.02	**0.378 ± 0.014**	0.316 ± 0.01
		10,000	0.376 ± 0.015	**0.38 ± 0.015**	**0.388 ± 0.012**	0.335 ± 0.015	**0.378 ± 0.017**	0.367 ± 0.017
		all	**0.384 ± 0.014**	0.383 ± 0.012	**0.408 ± 0.013**	0.348 ± 0.018	**0.383 ± 0.016**	0.354 ± 0.013
xenograft data (PDXs)	pre. onc.	0	–	0.715 ± 0.128	–	0.646 ± 0.171	–	0.7 ± 0.14
		10	0.243 ± 0.149	**0.366 ± 0.138**	0.196 ± 0.176	**0.308 ± 0.163**	**0.403 ± 0.138**	0.246 ± 0.149
		50	0.19 ± 0.18	**0.443 ± 0.13**	−0.131 ± 0.144	**0.558 ± 0.149** *	**0.541 ± 0.124**	0.424 ± 0.156
		all	0.502 ± 0.128	**0.592 ± 0.13**	0.334 ± 0.133	**0.353 ± 0.195**	**0.607 ± 0.123**	0.533 ± 0.124
	drug dev.	0	–	0.129 ± 0.057	–	−0.005 ± 0.049	–	−0.134 ± 0.08
		10	**−0.078 ± 0.136**	−0.254 ± 0.093	−0.133 ± 0.109	**0.06 ± 0.087**	−0.389 ± 0.028	**−0.255 ± 0.042**
		50	**−0.034 ± 0.068**	−0.418 ± 0.021	−0.171 ± 0.168	**−0.02 ± 0.099**	**−0.353 ± 0.088**	−0.451 ± 0.075
		all	**−0.251 ± 0.152**	−0.42 ± 0.135	**0.077 ± 0.141**	0.056 ± 0.098	−0.28 ± 0.112	**−0.267 ± 0.1**
Organoid data (PDOs)	pre. onc.	0	–	0.577 ± 0.021	–	0.488 ± 0.027	–	0.602 ± 0.028
		10	0.023 ± 0.036	**0.575 ± 0.048** *	−0.009 ± 0.021	**0.569 ± 0.031** *	−0.004 ± 0.02	**0.665 ± 0.029** *
		50	0.201 ± 0.036	**0.791 ± 0.013** *	0.066 ± 0.025	**0.691 ± 0.029** *	0.435 ± 0.038	**0.744 ± 0.021** *
		100	0.404 ± 0.047	**0.863 ± 0.013** *	0.077 ± 0.029	**0.77 ± 0.017** *	0.468 ± 0.079	**0.823 ± 0.014** *
		500	0.874 ± 0.015	**0.905 ± 0.007**	0.153 ± 0.028	**0.878 ± 0.008** *	**0.909 ± 0.007**	0.899 ± 0.008
		all	**0.909 ± 0.007**	0.904 ± 0.006	0.503 ± 0.076	**0.896 ± 0.008** *	**0.918 ± 0.007**	0.897 ± 0.007
	drug dev.	0	–	0.001 ± 0.037	–	−0.0 ± 0.041	–	−0.086 ± 0.033
		10	**−0.031 ± 0.041**	−0.06 ± 0.047	−0.033 ± 0.023	**−0.021 ± 0.036**	−0.053 ± 0.045	**−0.042 ± 0.035**
		50	0.002 ± 0.035	**0.107 ± 0.034**	**0.056 ± 0.025**	0.039 ± 0.026	0.108 ± 0.047	**0.205 ± 0.039**
		100	−0.035 ± 0.057	**0.241 ± 0.035** *	−0.028 ± 0.027	**0.03 ± 0.038**	0.181 ± 0.034	**0.225 ± 0.036**
		500	0.051 ± 0.046	**0.405 ± 0.027** *	0.003 ± 0.025	**0.177 ± 0.039** *	0.312 ± 0.063	**0.361 ± 0.022**
		all	0.282 ± 0.052	**0.44 ± 0.022** *	0.186 ± 0.044	**0.197 ± 0.038**	0.422 ± 0.029	**0.458 ± 0.031**

## Data Availability

The source code and the data to run the experiments used in this work are available at https://github.com/prassepaul/mlmed_transfer_learning, (accessed on 14 July 2022). All experiments reported in this paper are based on published data sets and have been downloaded from the respective data repositories or project websites. The GDSC data [3,4] are available at https://www.cancerrxgene.org, (accessed on 14 July 2022) and CCLE data [5] are available at the deepmap data portal CCLE database. Lung cancer xenografts data [16] have been obtained from the European Genome-phenome Archive (EGA) under the accession number EGAS00001002479. Pancreas PDO data [17] were downloaded from the NCBI dbGaP archive, accession number phs001611.v1.p1, and Beat AML data [15] were extracted from the supplementary tables of the publication.

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
