# Peer review of "Pre-Training on In Vitro and Fine-Tuning on Patient-Derived Data Improves Deep Neural Networks for Anti-Cancer Drug-Sensitivity Prediction"

_cancers, 2022, doi:10.3390/cancers14163950_

Round 1
Reviewer 1 Report
The presented manuscript is devoted to the problem of fine tuning neural networks for predicting drug effectiveness. The manuscript is well-written and seems like a completed research. However, several comments can be made. My major concern is that authors evaluate predictive performance as the Pearson correlation between predicted and measured drug sensitivity. High values of Pearson correlation does not mean good agreement between measurements, e.g. if one value is always two times greater than the other, the correlation will be perfect.There are also some minor points: - Order of model architectures presented on the Fig1 should correspond to the order they are introduced in the text. - Description of ConvNN architecture does not include what metric is being minimized - VENN diagram in the Fig2 caption should be spelled as ‘Venn’ - Pathway enrichment is not described in methods. Overall I suggest the manuscript requires minor revision.
Author Response
Response to Reviewer 1 Comments
Point 1: My major concern is that authors evaluate predictive performance as the Pearson correlation between predicted and measured drug sensitivity. High values of Pearson correlation does not mean good agreement between measurements, e.g. if one value is always two times greater than the other, the correlation will be perfect.
Response 1: We added a paragraph motivating the use of the Pearson correlation. The rationale is that in precision oncology, the goal is to identify the most promising drug candidates for a given cell line. Even if all predicted inhibitory concentrations are off by a constant factor, this goal is achieved when the predicted concentrations are perfectly correlated to the true concentrations. For drug design, the same argument applies because the goal is to identify the most promising cell lines for a given candidate drug compound. We agree that if our goal was to predict the therapy outcome, then the absolute value would be significant and the Pearson correlation would not be an appropriate metric.
Furthermore, we also added a small paragraph using the MSE as evaluation metric in the results section to address this issue.
Point 2: Order of model architectures presented on the Fig1 should correspond to the order they are introduced in the text
Response 2: We changed the order in Figure 1.
Point 3: Description of ConvNN architecture does not include what metric is being minimized.
Response 3: We added the training objective in the manuscript.
Point 4: VENN diagram in the Fig2 caption should be spelled as ‘Venn’.
Response 4: Thanks for spotting this spelling mistake. We changed it.
Point 5: Pathway enrichment is not described in methods.
Response 5: We added a paragraph describing how p-values for the pathway enrichment are calculated (see Section 2.5).
Reviewer 2 Report
In this manuscript, Prasse et al. aimed to pre-train deep-learning drug-sensitivity prediction models on a large data set under patients-centered construction. They explored a method for training pre-clinical anti-cancer drug-sensitivity models that are based on pre-training on in vitro drug-sensitivity data and fine-tuning on drug-sensitivity data based on patient-derived cell cultures and relevant models. The manuscript is well written and description and figures are clear and logical. However, there are only minor points that need to be addressed.
1- I am not familiar merit of pre-train deep-learning drug-sensitivity prediction models. The authors should add a flow chart to clarify this concern.
2- Statistical method description is vague. Please add pro and con of proposed approaches.
3- How the authors to validate the reliability of pre-train deep-learning drug-sensitivity prediction models in this study?
Author Response
Response to Reviewer 2 Comments
Point 1: I am not familiar merit of pre-train deep-learning drug-sensitivity prediction models. The authors should add a flow chart to clarify this concern.
Response 1: We added a new subfigure (A) in Figure 1 to show the overall Transfer Learning approach as a flow chart.
Point 2: Statistical method description is vague. Please add pro and con of proposed approaches.
Response 2: We added a motivation of using the Pearson correlation and pros and cons for this decision in Section 2.2.
Point 3: How the authors to validate the reliability of pre-train deep-learning drug-sensitivity prediction models in this study?
Response 3: Reliability on the one hand is assessed numerically by comparing predictions against the ground truth as shown in Table 2 of the manuscript. On the other hand, we assessed reliability of the predictions by exploring biological plausibility of the genes that gain high importance values by the models. Biological plausibility was judged by the strength of enrichment that the top-ranked genes achieve with respect to human pathways. We added a subsection (Section 2.5) addressing this issue. Additionally, we motivated the approach in a paragraph in the discussion section.